# Evaluation of the Efficiency of European Health Systems Using Fuzzy Data Envelopment Analysis

**DOI:** 10.3390/healthcare9101270

**Published:** 2021-09-26

**Authors:** Juan Cándido Gómez-Gallego, María Gómez-Gallego, Javier Fernando García-García, Ursula Faura-Martinez

**Affiliations:** 1Applied Economics Department, Faculty of Economics, University of Murcia, 30100 Murcia, Spain; 2Clinical Neuroscience Research Group, Faculty of Health Sciences, San Antonio Catholic University, 30107 Murcia, Spain; mggallego@ucam.edu (M.G.-G.); jfggar@gmail.com (J.F.G.-G.); 3Quantitative Methods Department, Faculty of Economics, University of Murcia, 30100 Murcia, Spain; faura@um.es

**Keywords:** efficiency, data envelopment analysis, fuzzy data, health system, income inequality, freedom economics

## Abstract

Many studies that assess efficiency in health systems are based on output mean values. That approach ignores the representativeness of the average statistic, which can become a serious problem in estimation. To solve this question, this research contributes in three different ways: the first aim is to evaluate the technical efficiency in the management of European health systems considering a set of DEA (Data Envelopment Analysis) and FDEA (Fuzzy Data Envelopment Analysis) models. A second goal is to assess the bias in the estimation of efficiency when applying the conventional DEA. The third objective is the evaluation of the statistical relationship between the bias in the efficiency estimation and the macroeconomic variables (income inequality and economic freedom). The main results show positive correlations between DEA and FDEA scores. Notwithstanding traditional DEA models overestimate efficiency scores. Furthermore, the size of the bias is positively related to income inequality and negative with economic freedom in the countries evaluated.

## 1. Introduction

Health expenditure (HE) has become a concern for most countries in recent years. HE represented around 10% of the Gross domestic product (GDP) for the 28 European Union (EU) countries as a whole in 2018: 11.47% in Germany, 11.26% in France, 5.56% in Romania, 6.7% in Hungary or 8.89% in Spain). In the present, these percentages have increased significantly due to the current coronavirus disease 2019 pandemic (COVID-19), as large amounts of resources are being invested in healthcare policies and prevention programmes to enhance the health of the population.

Health systems have three main goals: the health status of the population, the responsiveness, and equity [1]. The degree of achievement of these objectives is connected to the effectiveness and efficiency of the system as a whole. Performance measurements reveal if a particular health system is accomplishing its objectives.

Comparing health systems across countries can determine if they are achieving the desired results. Nowadays, datasets allowing the comparison of the performance and healthcare efficiency levels among different countries can be easily accessed [2]. Comparing socioeconomically similar countries (benchmarking) is one of the methods for measuring the performance and efficiency levels of health systems. Data-driven international comparison of health systems has long been considered important in guiding the decision makers in their discussions about health policies [3,4]. Although healthcare in each country is unique due to historical and socioeconomic differences, all health systems have a common function: provide accessible and effective health services to individuals at an optimum cost [5]. Therefore, countries can learn from each other through the comparison of their health system with others. The resulting conclusions on healthcare performance may strengthen the scientific basis of health policies applied at national and international levels [6].

There is extensive literature on the efficiency of health systems. Varabyova and Müller [7] performed a systematic review and meta-analysis of cross-country comparisons in OCDE (Organization for Cooperation and Economic Development) countries. The study of Asandului et al. [8] found a significant difference between the efficiency of public health systems in developed and developing European countries. Medeiros and Schwierz [9] also reported that there is considerable waste cost, which increases health expenditure, although the relative efficiency of health systems across participating EU (European Union) countries is different. Moreno et al. [10] assessed the efficiency of 29 EU countries in the year of the economic crisis, 2009, following the different productive approaches of the countries under study. Gearhart [11] analysed the impact of secondary environmental variables on OECD healthcare efficiency. Moreno-Serra et al. [12] evaluated the potential determinants of health system efficiency on Latin America and the Caribbean. Jordi et al. [13] assessed the efficiency of HS with a sample of 172 countries. Zhou et al. [14] analysed the healthcare efficiency of emerging countries. Other studies of interest on this issue are [15,16,17,18,19].

One of the most used methods to assess efficiency is the Data Envelopment Analysis (DEA), which measures the relative efficiencies of multiple decision-making units with similar goals and objectives [20,21]. A theoretical production frontier results from the efficiency scores of the countries whose health systems are considered efficient. On the other hand, countries situated below the frontier present inefficient health systems. This technique enables the establishment of a ranking of the efficiency of countries where each health system classified as inefficient has a reference set. As DEA output scores might be biased, and output and input variables might correlate, the use of bootstrapping techniques is recommended [22,23,24]. Bootstrap DEA allows better efficiency estimations because it analyses the sensitivity of measured efficiency scores to sampling variation. It reduces statistical uncertainty and measurement errors presented in original DEA estimates [25].

DEA models are limited to “crisp data”, that is, precise input and output values of the decision-making units (DMUs). Therefore, when the unit of analysis is a country’s health system as a whole, and when microdata is available for a given output, these data are generally summarised by calculating an average value. For example, it is a common practice to represent the life expectancy of a country in a year by estimating the average age distribution of the people who died that year. The main drawback of this approach is that these aggregate measures (statistics) ignore the possibility of important differences in the distribution forms of the output in each of the units evaluated, which might have some important implications in terms of inequality [26].

In this paper, we attempt to provide evidence on how efficiency measures estimated with DEA might be affected when variability in data is taken into account. To do this, we use fuzzy DEA (FDEA), which allows us to incorporate the hidden variability behind a mean (or crisp) value. This approach deals with imprecise data modelled through the notion of fuzziness [27,28] and usually consists in transforming the FDEA model into several conventional “crisp” DEA programs.

The literature devoted to FDEA has experienced a notable development in recent years [29,30,31,32,33]. Therefore, multiple alternative approaches deal with this type of analysis, such as [34,35,36,37] and more recently, [31,38,39]. Some recent contributions in healthcare area using this methodology are [40,41,42,43,44].

In particular, this paper applies the methodology proposed by [37] which is the most common and cited contribution to this field of research.

In this research, DEA, Bootstrap DEA, and FDEA are applied to evaluate the efficiency of the health systems of 28 European countries in 2017. The paper contributes in three different ways: the first aim is to evaluate the technical efficiency in the management of European health systems considering a set of DEA and FDEA models. A second goal is to assess the bias in the estimation of efficiency when applying the conventional DEA. The third objective is the examination of the statistical relationship between the bias in the efficiency estimation and the macroeconomic variables (income inequality, wealth per capita, and economic freedom).

The remainder of the paper is structured as follows: Section 2 introduces the necessary notation and background; in Section 3, the data and the variables used in the empirical analysis are presented; Section 4 includes the main results of the application of the Kao and Liu approach to the data of the countries evaluated and compares them with those obtained with more traditional DEA models; in Section 5, the conclusions are drawn.

## 2. Methodology

Fuzzy data envelopment analysis (FDEA) is a technique that allows to incorporate data variability into efficiency analysis, so that variables such as life expectancy (LE) and self-perceived health (SPH) are included in the DEA model as fuzzy data instead of a crisp data number. [30] carried out a review of the FDEA methods and establish a classification scheme with six categories, namely, the tolerance approach, the α-level-based approach, the fuzzy ranking approach, the possibility approach, the fuzzy arithmetic, and the fuzzy random/type-2 fuzzy set. Among them, the α-level-based approach is probably the most popular FDEA model, and the approach by [37] which belongs to this category, is the most cited and applied methodology.

The basic idea behind the Kao and Liu approach is to apply the α-cuts and Zadeh’s extension principle to transform a fuzzy DEA model into a series of conventional crisp DEA models. These conventional models are then solved by well-known linear programming techniques. In particular, Kao and Liu exploit the linear model (Equations (1)–(5)).
(1)$$Min\;\phi_{0}=\sum_{i=1}^{m}v_{i0}x_{i0}-\pi_{0}$$
s.t.
(2)$$\sum_{r=1}^{s}u_{r0}y_{r0}=1,$$
(3)$$\overset{m}{\underset{i=1}{\sum }}v_{i0}x_{ij}-\overset{s}{\underset{r=1}{\sum }}u_{r0}y_{rj}-\pi_{0}\;\geq 0\;\;\;\;\;\;\;j=1,\;2,\;\ldots .,\;n$$
(4)$$v_{i0\;}\geq 0\;\;\;\;i=1,\;2,\;\ldots ,\;m$$
(5)$$u_{r0\;}\geq 0\;\;\;\;r=1,\;2,\;\ldots ,\;s$$

Let us suppose that the inputs ${\tilde{x}}_{ij}$, j=1,…,n, i=1,…,m, and outputs ${\tilde{y}}_{rj}$, j=1,…,n, r=1,…,s, are fuzzy numbers with membership functions $\mu_{{\tilde{x}}_{ij}{}_{}}$ and $\mu_{{\tilde{y}}_{rj}}$, respectively. The membership functions characterize the fuzzy numbers and measures the degree of truth of the statement that the represented magnitude takes a specific.

Particularly, we use a special type of trapezoidal fuzzy number with a membership function $\mu_{\tilde{A}}$ expressed as follows [45]:(6)$$\mu_{\tilde{A}}\left(z\right)=\{\begin{array}{cc} \frac{z-a^{l}}{a^{m_{1}}-a^{l}}, & a^{l}\leq z\leq a^{m_{1}}\\ 1, & a^{m_{1}}\leq z\leq a^{m_{2}}\\ \frac{a^{u}-z}{a^{u}-a^{m_{2}}}, & a^{m_{2}}\leq z\leq a^{u}\\ 0, & Otherwise\; \end{array}$$

The trapezoidal fuzzy number $\tilde{A}=\left(a^{l},a^{m_{1}},a^{m_{2}},a^{u}\right)$ is reduced to a real number A, if $a^{l}=a^{m_{1}}=a^{m_{2}}=a^{u}$. Conversely, a real number *A* can be written as a trapezoidal fuzzy number $\tilde{A}=\left(a,a,a,a\right).$ If $a^{m}=a^{m_{1}}=a^{m_{2}}$, then $\tilde{A}=\left(a^{l},a^{m},a^{u}\right)$ is called a triangular fuzzy number. We assume that all fuzzy number used throughout the paper are trapezoidal fuzzy number.

A usual notion in fuzzy number theory is the α-cuts, also called α-possibility level sets, which are derived from the membership function of the membership function, being ${\left({\tilde{x}}_{ij}\right)}_{\alpha }=\left\{z:\;\mu_{{\tilde{x}}_{ij}}\left(z\right)\geq \alpha \right\}$ and ${\left({\tilde{y}}_{rj}\right)}_{\alpha }=\left\{z:\;\mu_{{\tilde{y}}_{rj}}\left(z\right)\geq \alpha \right\}$ the α-cuts of ${\tilde{x}}_{ij}$ y ${\tilde{y}}_{rj}$, respectively. Each α-cut generates a confidence interval for the represented magnitude (input or output). We generally represent the α-cuts by means of their highest and lowest values in the resulting interval: ${\left({\tilde{x}}_{ij}\right)}_{\alpha }=\left[{\left({\tilde{x}}_{ij}\right)}_{\alpha }^{L},{\left({\tilde{x}}_{ij}\right)}_{\alpha }^{U}\;\right]$ and ${\left({\tilde{y}}_{rj}\right)}_{\alpha }=\left[{\left({\tilde{y}}_{rj}\right)}_{\alpha }^{L},{\left({\tilde{y}}_{rj}\right)}_{\alpha }^{U}\;\right]$.

Without loss of generality, it is assumed that all input and output data for all DMUs are fuzzy numbers, since crisp values may be represented by degenerated membership functions that only have one value in their domain.

In this way, the radial DEA model expressed by means of the input and output fuzzy numbers would be formulated as follows (Equations (7)–(11)):(7)$$Min{\tilde{\phi }}_{0}^{*}=\sum_{i=1}^{m}v_{i0}{\tilde{x}}_{i0}-\pi_{0}$$
s.t.
(8)$$\sum_{r=1}^{s}u_{r0}{\tilde{y}}_{r0}=1,$$
(9)$$\overset{m}{\underset{i=1}{\sum }}v_{i0}{\tilde{x}}_{ij}-\overset{s}{\underset{r=1}{\sum }}u_{r0}{\tilde{y}}_{rj}-\pi_{0}\;\geq 0\;\;\;\;\;\;\;j=1,\;2,\;\ldots .,\;n$$
(10)$$v_{i0\;}\geq 0\;\;\;i=1,\;2,\;\ldots ,\;n$$
(11)$$u_{r0\;}\geq 0\;\;\;\;r=1,\;2,\;\ldots ,\;s$$
where ${\tilde{\phi }}_{0}^{*}$ represents the efficiency score for DMU_0_ and that is also a fuzzy number with a certain membership function. The Kao and Liu approach is based on determining this membership function from different α-cuts of the fuzzy numbers that appear in model (Equations (7)–(11)). Indeed, given a certain possibility level α (0<α≤1), it is possible to determine the lowest value of the corresponding α-cut for the membership function of ${\tilde{\phi }}_{0}^{*}$ through (Equations (12)–(17)).
(12)$${\left({\tilde{\phi }}_{0}^{*}\right)}_{\alpha }^{L}=Min\;\overset{m}{\underset{i=1}{\sum }}v_{i0}{\left({\tilde{x}}_{i0}\right)}_{\alpha }^{U}-\pi_{0}$$
s.t.
(13)$$\sum_{r=1}^{s}u_{r0}{\left({\tilde{y}}_{r0}\right)}_{\alpha }^{L}=1,$$
(14)$$\overset{m}{\underset{i=1}{\sum }}v_{i0}{\left({\tilde{x}}_{i0}\right)}_{\alpha }^{L}-\overset{s}{\underset{r=1}{\sum }}u_{r0}{\left({\tilde{y}}_{r0}\right)}_{\alpha }^{U}-\pi_{0}\;\geq 0\;\;\;\;\;\;\;j\neq 0$$
(15)$$\overset{m}{\underset{i=1}{\sum }}v_{i0}{\left({\tilde{x}}_{i0}\right)}_{\alpha }^{U}-\overset{s}{\underset{r=1}{\sum }}u_{r0}{\left({\tilde{y}}_{r0}\right)}_{\alpha }^{L}-\pi_{0}\;\geq 0\;\;\;\;\;\;\;$$
(16)$$v_{i0\;}\geq 0\;\;\;\;i=1,\;2,\;\ldots ,\;m$$
(17)$$u_{r0\;}\geq 0\;\;\;\;r=1,\;2,\;\ldots ,\;s$$

The idea is to calculate the smallest efficiency score of DMU_0_ compared with the other *n*−1 DMUs. To that end, we must set the output level of DMU_0_ and the input levels of all other units to their lowest values and the other set the input level of DMU_0_ and the output levels of all other DMUs to their highest values.

By analogy, it is possible to calculate the highest value of the corresponding α-cut for the membership function of ${\tilde{\phi }}_{0}^{*}$ as follows (Equations (18)–(23)).
(18)$${\left({\tilde{\phi }}_{0}^{*}\right)}_{\alpha }^{U}=Min\;\overset{m}{\underset{i=1}{\sum }}v_{i0}{\left({\tilde{x}}_{i0}\right)}_{\alpha }^{L}-\pi_{0}$$
s.t.
(19)$$\sum_{r=1}^{s}u_{r0}{\left({\tilde{y}}_{r0}\right)}_{\alpha }^{U}=1,$$
(20)$$\overset{m}{\underset{i=1}{\sum }}v_{i0}{\left({\tilde{x}}_{i0}\right)}_{\alpha }^{U}-\overset{s}{\underset{r=1}{\sum }}u_{r0}{\left({\tilde{y}}_{r0}\right)}_{\alpha }^{L}-\pi_{0}\;\geq 0\;\;\;\;\;\;\;j\neq 0$$
(21)$$\overset{m}{\underset{i=1}{\sum }}v_{i0}{\left({\tilde{x}}_{i0}\right)}_{\alpha }^{L}-\overset{s}{\underset{r=1}{\sum }}u_{r0}{\left({\tilde{y}}_{r0}\right)}_{\alpha }^{U}-\pi_{0}\;\geq 0\;\;\;\;\;\;\;$$
(22)$$v_{i0\;}\geq 0\;\;\;\;i=1,\;2,\;\ldots ,\;m$$
(23)$$u_{r0\;}\geq 0\;\;\;\;r=1,\;2,\;\ldots ,\;s$$

Linear programs (Equations (12)–(23)) permit the systematic study of the form of the membership function of the “fuzzy” efficiency score of DMU_0_ simply by determining the interval $\left[{\left({\tilde{\phi }}_{0}^{*}\right)}_{\alpha }^{L},{\left({\tilde{\phi }}_{0}^{*}\right)}_{\alpha }^{U}\right]$ for different values of α (0<α≤1).

After the fuzzy efficiency scores are determined for all DMUs in the sample, an interesting procedure is to rank the units to determine the better ones. Although there are several methods for ranking units in FDEA, we will resort to that suggested by [36] since it is based upon the α-cuts y based on [46]. The proposed index is defined as:(24)I0=[∑k=0h((ϕ˜0∗)αkU−c)][∑k=0h((ϕ˜0∗)αkU−c)−∑k=0h((ϕ˜0∗)αkL−d)]
where $\;c=min_{j,k}\left\{{\left({\tilde{\phi }}_{j}^{*}\right)}_{\alpha_{k}}^{L}\right\}$ y $\;d=max_{j,k}\left\{{\left({\tilde{\phi }}_{j}^{*}\right)}_{\alpha_{k}}^{U}\right\}$.

## 3. Data and Variables

In this paper, the production function has been defined from three inputs (total health expenditure, total number of doctors, and total number of beds) and three outputs (life expectancy, child survival ratio, and self-perceived health). The data source is Eurostat and the data refer to the 28 EU countries for the year 2017. According to Eurostat, the data respond to the following definitions:

Inputs:Health expenditure (HE). It measures the final consumption of health care goods and services including personal health care (curative care, rehabilitative care, long-term care, ancillary services, and medical goods) and collective services (prevention and public health services as well as health).Physicians (P). They apply preventive and curative measures, improve or develop concepts, theories, and operational methods and conduct research in the area of medicine and health care. Physicians may be counted according to different concepts such as “practicing”, “professionally active” or “licensed to practice”. Practicing physicians provide services directly to patients.Hospital beds (B). It provides information on health care capacities, i.e., on the maximum number of patients who can be treated by hospitals. Total hospital beds are all hospital beds which are regularly maintained and staffed and immediately available for the care of admitted patients; both occupied and unoccupied beds are covered. Hospitals are defined according to the classification of health care providers of the System of Health Accounts (SHA); all public and private hospitals should be covered.Life expectancy at birth (LE). It is defined as the mean number of years that a new-born child can expect to live if subjected throughout his life to the current mortality conditions (age specific probabilities of dying).Infant Survival Rate (ISR). It is defined as infant survival per thousand live births.Self-perceived health (SPH). It expresses subjective assessment by the respondent of his/her health. Indicators based on this concept can be used to evaluate the general health status, health inequalities, and health care needs at the population level.

In order to explain the differences between the efficiency estimates resulting from the FDEA and the standard DEA other variables considered in the study [12,14,47,48,49]:The Gini coefficient of equivalised disposable income inequality (GC) is defined as the relationship of cumulative shares of the population arranged according to the level of equivalised disposable income, to the cumulative share of the equivalised total disposable income received by them. It takes values between 0 and 100 (the higher the index, the greater the inequality in disposable income).Gross domestic product per capita (GDPpc) is a measure for the economic activity. It is defined as the value of all goods and services produced less the value of any goods or services used in their creation. The volume index of GDP per capita in Purchasing Power Standards (PPS) is expressed in relation to the European Union average set to equal 100 (it can also be expressed in Euros).Economic Freedom (EF) (obtained from Heritage Foundation). In an economically free society, individuals are free to work, produce, consume, and invest in any way they please. Governments allow labour, capital, and goods to move freely, and refrain from coercion or constraint of liberty beyond the extent necessary to protect. The score is obtained by the aggregation of the scores in four dimensions: the rule of law (RL), government size (GS), regulatory efficiency (RE), and open markets (OM).

In this paper, the data related to the variables HE, P, B, and ISR have been considered as crisp data, whereas the observed data of the LE and SPH outputs are considered fuzzy data. The variable LE is associated with the uncertainty of the concept of a statistic whose representativeness can be partly approximated by characteristics such as the variance, skewness, and kurtosis of its distribution. To apply the Kao and Liu approach [37], the value of the variable for each country is modelled as a fuzzy number. To do this, we estimate a kernel function from the data corresponding to the variable LE for each country. In particular, we use the “Kernel density estimation” [50,51] function of the Stata 13.0 software (College Station, TX, USA). Figure 1 shows the kernel density estimate for Spain, as an illustrating example of this.

In this case (Spain), the following statistics are obtained: mean (82.73), SD (12.66), Skewness coefficient (−1.54), and kurtosis coefficient (3.93). Following the Kao and Liu procedure, given a certain level of possibility, we need to determine the α-cut (an interval, which can be described by its lowest and highest values). For Spain, the α-cut could be defined from the membership function:(25)$$\mu_{{\tilde{y}}_{rj}}\left(z\right)=\{\begin{array}{cc} \frac{z-70}{80-70}, & 70\leq z\leq 80\\ 1, & 80\leq z\leq 85\\ \frac{95-z}{95-85}, & 85\leq z\leq 95\;\\ 0, & \mathrm{otherwise} \end{array}$$

The case of the self-perceived health variable is different, as the information is obtained through questionnaires, where the perception that the person has about their health has been measured as linguistic variables [45,52,53]. To measure people’s perception, five linguistic variables are used: “very bad (VB)”, “bad (B)”, “fair (F)”, “good (G)”, and “very good (VG)”. These linguistic terms are distributed on a scale [0, 1] as indicated in Table 1. For each country, the frequency of respondents for each linguistic variable is known in order to construct the corresponding assessment intervals.

The SPH fuzzy values are calculated from the trapezoidal fuzzy numbers for each health system as follows:(26)$$\begin{array}{ll} \left[l_{i},\;m_{1i},m_{2i},\;u_{i}\right] & \\ & =f_{i1}*\left[0.8,\;0.9,\;0.9,\;1\right]+f_{i2}*\left[0.6,\;0.7,\;0.7,\;0.8\right]+f_{i3}\\ & *\left[0.4,\;0.5,\;0.5,\;0.6\right]+f_{i14}*\left[0.2,\;0.3,\;0.3,\;0.4\right]\\ & +f_{i5}\left[0,\;0.1,\;0.1,\;0.2\right] \end{array}$$
where $f_{ij\;}\left(i=1,\;2,\;\cdots ,28;\;j=VG,\;G,\;F,\;B\;y\;VB\right)$ is the frecuency or respondents corresponding to the i-th country and the j-th variable.

Table 2 shows the descriptive statistics of input and output variables together with the covariates, indicating if they were considered crisp data or fuzzy data.

## 4. Results

In this section, we present the results obtained when the DEA and FDEA models are applied to the data available for the sample of the 28 EU countries.

Specifically, we have estimated efficiency scores using three alternative approaches. First, we calculate a standard DEA using the mean values of all the variables, i.e., treating LE and SPH as crisp. Second, we apply the bootstrap procedure suggested by [23] to the same mean values. This approach allows us to obtain bias-corrected efficiency scores and their reference confidence intervals (the estimations were performed using the package “Benchmarking” [54] in R with 1000 replications). Finally, we obtain fuzzy efficiency estimates using the approach by Kao and Liu (2000) for different α-cuts (α = 0.2, 0.4, 0.6, 0.8, 0.9 and 1) incorporating LE and SPH as fuzzy numbers. For each α, we present both the lowest and the highest value of the confidence interval for the fuzzy efficiency score; therefore, we have interval $\left[{\left({\tilde{\phi }}_{0}^{*}\right)}_{\alpha }^{L},{\left({\tilde{\phi }}_{0}^{*}\right)}_{\alpha }^{U}\right]$. Moreover, we calculate the index I_0_ (Equation (5)), which reflects a summary of the different results determined for the set of α’s. In all the estimations we assume variable returns to scale and an output orientation, since we consider that countries are always attempting to maximize the health level and cannot easily reduce their inputs, at least in the short term.

From our results (Table 3), we observe that the correlation of the scores estimated with the standard DEA and DEA-BC models with the upper end of the interval is high and very high. However, the correlations with the scores estimated with FDEA for α=1 are lower, although higher with the values that define the upper end of the interval. The correlations between the DEA and DEA-BC scores with the $I_{0}$ index are moderate (somewhat higher than 0.7) and are always lower than the respective ones with the upper end of the FDEA scores.

Table 4 shows the results relative to the efficiency scores estimated by applying the DEA, DEA-BC and FDEA models for different values of α-cut and the Kao and Liu index.

In the first place, when comparing the DEA and DEA-BC efficiency scores, it is observed that the bias is always positive (mean = 0.064; dt = 0.03) in Cyprus, Poland, Portugal, Romania, and Sweden, with an overestimation of the efficiency greater than 10%. In addition, the table shows that these countries belong to the group that sets the efficient frontier; therefore, the overestimation bias affects the efficiency scores for the total units of the sample evaluated.

When comparing the DEA scores with the averages of the FDEA scores for α = 1, it is found that the standard DEA model overestimates the efficiency score for a group of countries (Bulgaria, Croatia, Greece, Hungary, Italy, Latvia, Portugal, Slovakia, Spain, and the United Kingdom). Specifically, for Croatia, Greece, and Portugal, the overestimation bias is higher than 10% and 7% for Bulgaria. This group of countries has an income inequality index higher than the EU-28 average (Bulgaria (40.2), Greece (33.3), Italy (32.7), Latvia (34.5), Portugal (33.5), Spain (34.1), the United Kingdom (33.1), the EU (30)). They also have a GDP per capita lower than the EU average (Bulgaria (7400), Croatia (11,920), Greece (16,470), Hungary (12,970), Italy (28,690), Latvia (13,890), Portugal (19,020), Slovakia (15,290), Spain (24,970), the EU (29,440)). Additionally, they have a PE index lower than the EU-28 average (Bulgaria (67.90), Croatia (59.40), Greece (55), Hungary (65.80), Italy (62.50), Portugal (62.70), Slovakia (65.70), Spain (63.60), the EU (69.44)).

Furthermore, Table 4 shows that, according to the standard DEA, nine countries are evaluated as efficient (Croatia, Cyprus, Ireland, Poland, Portugal, Romania, Spain, Sweden, and the United Kingdom). However, for FDEA and α = 1, neither Croatia nor Spain are efficient and only two of the nine (Poland and Sweden) are identified as efficient in the FDEA model for any value of α. Therefore, Spain, for example, should improve its outputs by 29% according to the value of I_0_, given its input levels, notwithstanding the fact that it is deemed efficient by the standard DEA. The DEA-BC model suggests that this country should improve its results by around 14%.

Unlike the standard DEA, the application of the FDEA and the Kao and Liu index allow us to greatly overcome the problem of the ordering of the evaluated units. The I_0_ index allows to establish an almost complete ranking of countries (with the exception of Poland and Sweden) and thus facilitate the identification of best practices. In fact, the Spearman correlations between the ordering according to the I_0_ values and according to the DEA scores is moderate (0.605, *p* < 0.01).

In addition, Table 4 shows that there is an overestimation bias when comparing the standard DEA efficiency scores with the respective values of the Kao and Liu. It represents almost seventeen percentage points on average, so it should be taken into account in the management strategies of the evaluated health systems. Likewise, an overestimation effect of more than ten percentage points results when the DEA-BC scores are compared with the values of the Kao and Liu index.

Therefore, the results shown in Table 4 reveal important differences between the efficiency scores estimated when applying the DEA and FDEA models. These differences are caused by the different forms that the distributions of the LE and SPH variables have in each of the 28 EU countries. This result is in line with Aparicio et al. (2019). They provided evidence that the underestimation of inefficiencies by traditional DEA models is due to the fact that these models ignore the different variability of the fuzzy outputs considered in the DMUs.

Now, what characteristics of the countries account for the different variability in the microdata of the LE and SPH variables? Can income inequality be considered a cause of variability in the age of mortality? What about the variability of self-perceived health? Similar questions can be asked regarding the GDP per capita and economic freedom variables.

To give an answer to such questions and in accordance with the objectives of this work, the relationship between the magnitude of the overestimation bias of the DEA efficiency and the three macroeconomic characteristics (GC, GDPpc, and EF with its dimensions: RL, GS, RE, and OM) has been studied. The results in Table 5 show that the bias is greater for those countries with the highest coefficient of income inequality. Likewise, less economic freedom is associated with greater efficiency overestimation biases. The relationship between the bias and the economic freedom is maintained between bias and each of the four dimensions of freedom, although with different size in effect and significance. The standardised coefficients indicate similar sizes of the effects of both explanatory variables. As Table 5 indicates, the explanatory power of the CG and EF variables is quantitatively very important, which should constitute an alert when carrying out and interpreting studies on the performance of health systems.

In this section, we must state that the results obtained when including the GDPpc as an explanatory variable are not clear; in fact, they interfere by mediating the relationships between the CG and EF variables with the DEA estimation bias.

Our results are consistent with those of other studies that have attempted to establish a relationship between income inequality and population health status [55,56].

The fact that there is a high degree of income inequality in a developed country can affect health in two different ways. Firstly, because it implies that a substantial segment of the population is impoverished, and poverty affects health negatively. Secondly, the aggregate level of inequality may affect the population in general, as there may be higher levels of psychosocial stress, resulting from comparisons between individuals with different levels of purchasing power [57,58], and more precarious conditions of tangible material [59], which indirectly also damages cohesion and erodes social welfare. The most unequal societies have a wide range of social problems that affect physical and mental health, education, violence, incarceration, or social immobility to the extent that people living in disadvantaged circumstances have worse health, more disabilities, and shorter lives than those who are wealthier [47,48]. [59] conclude that a large number of studies find that individuals living in regions with higher rates of wealth inequality have a higher risk of presenting health problems and have a higher incidence of premature death, regardless of sex, age, and socioeconomic status.

Similarly, a strong negative association is found between the income inequality indicator and the self-perceived health measure [60]. Our results are consistent with these findings, since the relationship between poverty and income inequality with the positive bias in the estimation of efficiency could be due to the variability in the outcome measures considered (life expectancy and self-perceived health), which reflect inequalities in people’s health.

## 5. Conclusions

The fuzzy DEA approach developed by Kao and Liu was applied to evaluate the performance of a sample of European countries in relation to the management of their health systems. This methodology allows us to take into account the variability in some variables such as life expectancy or perceived quality of life, considered as outputs of health systems, which, frequently, are overlooked in empirical analysis that evaluate the performance of health systems.

Our results indicate that the estimated performance measures of health systems, obtained with the FDEA approach, present significant levels of correlation with efficiency scores calculated with traditional DEA models. Consequently, we could rely on the validity of traditional measures, but it is worth mentioning that we find some relevant divergences in efficiency scores obtained with those methods for some countries. Specifically, we note that the traditional DEA tends to overestimate the level of efficiency with respect to the measures obtained through FDEA.

In addition, relevant divergences have been observed between the DEA and FDEA models according to the identification of efficient countries. We note that only twenty-two percent of the efficient countries under the standard model are also identified as efficient in the FDEA model for any level of α. This finding is particularly relevant since it affects the efficiency scores of all the countries. If these countries are not properly identified, it is difficult to establish guidelines for the rest. Therefore, according to our findings, we must be cautious when interpreting estimated efficiency measures using aggregate data for countries, as they may not provide a precise measure of its real performance. According to estimated regression models, it has been found, as explanations of overestimation bias, certain economic variables, such as per capita income, inequality in available income and economic freedom.

Finally, we want to point out that the correlation between, for example, income inequality and mortality age could be weaker in samples from countries with well-developed welfare systems [49]. In northern European and continental European countries, the level of development can help cushion the adverse effect of health impacts and, ultimately, variability in mortality. The factors cushioning the effect may be of historical, social, or cultural nature, and may be associated with both the hierarchical nature of societies, as indicated by income inequality, and with the health of the population. In this line, further research is required to help establish with greater precision the true interrelationships between the multiple economic, social, and health variables that take part in the productive process of health systems. That is our next goal.

## Figures and Tables

**Figure 1 healthcare-09-01270-f001:**
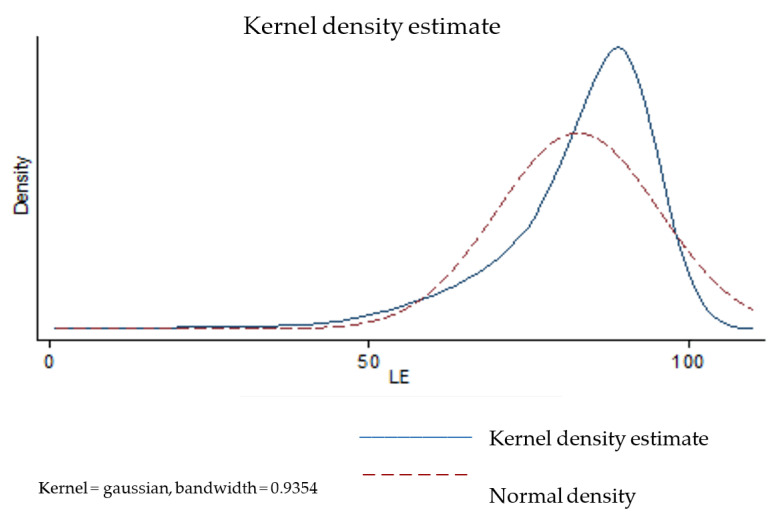
Kernel density estimate for Spain.

**Table 1 healthcare-09-01270-t001:** The linguistic variables and trapezoidal fuzzy numbers.

Linguistic Variable	Trapezoidal Fuzzy Number
Very good	[0.8, 0.9, 0.9, 1]
Good	[0.6, 0.7, 0.7, 0.8]
Fair	[0.4, 0.5, 0.5, 0.6]
Bad	[0.2, 0.3, 0.3, 0.4]
Very bad	[0, 0.1, 0.1, 0.2]

**Table 2 healthcare-09-01270-t002:** Descriptive statistics of variables for the sample of countries.

Variable			Mean	Std. Dev.	Min.	Max.
Outputs	(Fuzzy)	LE	80.09	2.70	71.96	84.24
		SPH	65.92	4.68	45.54	84.78
	(Crisp)	ISR	309.50	90.39	148.25	499.00
Inputs	(Crisp)	HE	2474.40	1008.80	1029.15	4299.83
		P	362.58	78.84	236.00	610.00
		B	491.20	170.94	222.49	800.23
Covariates	(Crisp)	GC	30.03	4.03	23.20	40.20
		GDPpc	29,430.00	19,271.65	7390.00	95,170.00
		FE	69.44	8.91	48.02	88.58
		RL	61.55	17.68	30.73	92.16
		GS	65.36	17.10	35.54	90.91
		RE	72.62	8.06	59.42	90.05
		OM	71.22	13.80	36.04	85.66

LE: life expectancy at birth; SPH: self-perceived health; ISR: infant survival rate; HE: health expenditure; P: physicians; B: beds; GC: Gini coefficient of equivalized disposable income; GDPpc: gross domestic product per capita; EF: economic freedom; RL: rule of law; GS: government size; RE: regulatory efficiency and OM: open markets.

**Table 3 healthcare-09-01270-t003:** Correlation coefficients among different approaches.

	DEA	DEA-BC	L-DEA-BC	U-DEA-BC	L-FDEA (α = 1)	U-FDEA (α = 1)
DEA						
DEA-BC	0.963 **					
L-DEA-BC	0.888 **	0.973 **				
U-DEA-BC	0.987 **	0.963 **	0.888 **			
L-FDEA (α = 1)	0.693 **	0.626 **	0.547 **	0.692 **		
U-FDEA (α = 1)	0.770 **	0.778 **	0.747 **	0.769 **	0.861 **	
$I_{0}$	0.721 **	0.650 **	0.571 **	0.720 **	0.992 **	0.862 **

** Correlation is significant at *p* < 0.001. DEA: Data Envelopment Analysis; DEA-BC: Data Envelopment Analysis bias corrected; L-DEA-BC: lower end of the confidence interval of bias-corrected DEA scores; U-DEA-BC: upper end of the confidence interval of bias-corrected DEA scores; L-FDEA: lower end of the interval of fuzzy DEA scores; U-FDEA: upper end of the interval of fuzzy DEA scores.

**Table 4 healthcare-09-01270-t004:** Efficiency scores according to the different models and α-cuts.

Country	DEA	DEA Bootstrap				FDEA (Different α-Cuts)										
		Bias	Mean	C. Interval		0.2		0.4		0.6		0.8		1		
	Scores	Scores	Scores	2.5%	97.5%	E^L^	E^U^	E^L^	E^U^	E^L^	E^U^	E^L^	E^U^	E^L^	E^U^	I_j_
Austria	0.560	0.024	0.536	0.515	0.558	0.510	0.730	0.520	0.720	0.530	0.710	0.540	0.690	0.560	0.680	0.295
Belgium	0.886	0.028	0.858	0.826	0.882	0.730	1.000	0.740	1.000	0.760	1.000	0.780	0.990	0.790	0.970	0.668
Bulgaria	0.839	0.055	0.784	0.716	0.836	0.640	0.920	0.650	0.900	0.670	0.880	0.680	0.860	0.690	0.850	0.527
Croatia	1.000	0.049	0.951	0.911	0.996	0.740	1.000	0.750	1.000	0.770	1.000	0.790	1.000	0.800	0.980	0.679
Cyprus	1.000	0.158	0.842	0.704	0.996	0.920	1.000	0.940	1.000	0.960	1.000	0.980	1.000	1.000	1.000	0.925
Czechia	0.749	0.044	0.706	0.672	0.746	0.780	1.000	0.790	1.000	0.810	1.000	0.820	1.000	0.840	1.000	0.718
Denmark	0.905	0.059	0.846	0.788	0.902	0.770	1.000	0.790	1.000	0.810	1.000	0.820	1.000	0.840	1.000	0.716
Estonia	0.951	0.054	0.897	0.834	0.947	0.960	1.000	0.980	1.000	1.000	1.000	1.000	1.000	1.000	1.000	0.976
Finland	0.950	0.067	0.883	0.819	0.946	0.910	1.000	0.930	1.000	0.940	1.000	0.960	1.000	0.980	1.000	0.897
France	0.833	0.032	0.801	0.766	0.830	0.710	1.000	0.720	0.990	0.730	0.970	0.750	0.950	0.760	0.930	0.633
Germany	0.606	0.028	0.578	0.554	0.603	0.530	0.760	0.540	0.740	0.550	0.730	0.560	0.710	0.570	0.700	0.326
Greece	0.964	0.065	0.899	0.817	0.961	0.670	0.960	0.680	0.940	0.700	0.920	0.710	0.900	0.730	0.890	0.577
Hungary	0.861	0.055	0.806	0.767	0.857	0.680	0.970	0.690	0.950	0.710	0.940	0.720	0.920	0.740	0.900	0.593
Ireland	1.000	0.061	0.939	0.866	0.996	0.950	1.000	0.970	1.000	0.990	1.000	1.000	1.000	1.000	1.000	0.965
Italy	0.940	0.049	0.891	0.839	0.935	0.730	1.000	0.750	1.000	0.760	1.000	0.780	0.990	0.790	0.970	0.669
Latvia	0.991	0.046	0.945	0.902	0.987	0.790	1.000	0.810	1.000	0.830	1.000	0.840	1.000	0.860	1.000	0.738
Lithuania	0.791	0.039	0.752	0.709	0.787	0.680	0.970	0.690	0.960	0.710	0.940	0.720	0.920	0.740	0.900	0.594
Luxembourg	0.896	0.038	0.858	0.824	0.891	0.890	1.000	0.910	1.000	0.930	1.000	0.950	1.000	0.970	1.000	0.875
Malta	0.791	0.044	0.748	0.714	0.788	0.840	1.000	0.860	1.000	0.870	1.000	0.890	1.000	0.910	1.000	0.795
Netherlands	0.790	0.054	0.736	0.676	0.787	0.680	0.970	0.690	0.950	0.710	0.940	0.720	0.920	0.740	0.900	0.593
Poland	1.000	0.101	0.899	0.843	0.996	1.000	1.000	1.000	1.000	1.000	1.000	1.000	1.000	1.000	1.000	1.000
Portugal	1.000	0.119	0.881	0.819	0.996	0.670	0.960	0.680	0.940	0.700	0.920	0.710	0.900	0.730	0.890	0.577
Romania	1.000	0.129	0.871	0.790	0.995	0.990	1.000	1.000	1.000	1.000	1.000	1.000	1.000	1.000	1.000	0.996
Slovakia	0.876	0.049	0.827	0.791	0.872	0.680	0.980	0.700	0.960	0.710	0.940	0.730	0.920	0.740	0.910	0.600
Slovenia	0.924	0.069	0.854	0.797	0.921	0.920	1.000	0.940	1.000	0.960	1.000	0.980	1.000	1.000	1.000	0.925
Spain	1.000	0.051	0.949	0.894	0.996	0.780	1.000	0.790	1.000	0.810	1.000	0.820	1.000	0.840	1.000	0.718
Sweden	1.000	0.136	0.864	0.786	0.995	1.000	1.000	1.000	1.000	1.000	1.000	1.000	1.000	1.000	1.000	1.000
United K.	1.000	0.084	0.916	0.844	0.996	0.870	1.000	0.890	1.000	0.910	1.000	0.930	1.000	0.940	1.000	0.842
Mean	0.897	0.064	0.833	0.778	0.893	0.786	0.972	0.800	0.966	0.815	0.960	0.828	0.953	0.841	0.945	0.729

DEA: Data Envelopment Analysis; FDEA: Fuzzy Data Envelopment Analysis. L: lower. U: upper.

**Table 5 healthcare-09-01270-t005:** Linear regression result: DEA overestimation bias.

		Non-Standardized Coefficients		Standardized Coefficients	*t*	*P* >|*t*|	$R_{ajust}^{2}$
		B	Std. Err.	Beta			
1	constant	0.236	0.104		2.270	0.031	0.443
	GC	0.010	0.003	0.482	3.360	0.003	
	EF	−0.006	0.002	−0.500	−3.481	0.002	
2	constant	−0.254	0.109		−2.330	0.029	0.476
	GC	0.012	0.003	0.586	3.932	0.001	
	GS	−0.001	0.000	−0.523	−3.532	0.002	
	RL	−0.001	0.000	−0.308	−2.145	0.042	

GC: Gini coefficient of equivalized disposable income; EF: economic freedom; RL: rule of law; GS: government size.

## Data Availability

Data are publicly available from Eurostat database, and may be shared directly upon request.

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
