# Peer review of "Evaluation of the Efficiency of European Health Systems Using Fuzzy Data Envelopment Analysis"

_healthcare, 2021, doi:10.3390/healthcare9101270_

Round 1

Reviewer 1 Report

It has been well modified. I think that it can be accepted now.

Reviewer 2 Report

Thank you for the revision. I just have two minor comments:

Citations with JC Gomez-Gallego are using an hyphen, but the authors name do not. Maybe they are not identical, though.

I am not convinced of the usefulness of Table 4.
- Is there a graphical representation possible?
- At least the Mean row should be highlighted/separated. Potentially a median row could be added.
- ISO Country codes could be used. At least United K. should be labeled UK
- Potentially different column widt so that E^L and E^U for the same cut are closer together, or an interval notation.

This manuscript is a resubmission of an earlier submission. The following is a list of the peer review reports and author responses from that submission.

Round 1

Reviewer 1 Report

Comments / Questions

  • Overall, the paper is very well readable. Please check however again for language mistakes (p. 3: "En este paper we asume", "frecuency")
  • I think the references to the equations are wrong (i.e., each LP got multiple equation numbers, but the text refers only to the LP numbers) 
  • The LPs refer to models only with fuzzy data, but the model set-up includes two non-crisp factors. I suggest to provide the LPs that were actually solved to obtain your results
  • Please provide which estimator has been used in the kernel density estimation
  • Please provide a short clarification for the reader on how the alpha-cut for Spain was selected and why it is skewed around the mean
  • I am not sure if I understood correctly: Are wages for physicians contained in the health expenditure data? If so, is the double counting a problem?
  • P9: If I understand correctly, the shift in the Simar Wilson bias correction is basically by construction because it assumes an underlying downward bias of the frontier estimate. 
  • P9: What is the source of the overestimation of the efficiency scores? Is there a relationship between dimensionality of the LP and the overestimation? 
  • Table 4 is not reader-friendly. A simple reformatting might help the reader. An additional line for sample means might be useful 
  • You may need additional descriptive statistics, because it is really hard to somehow understand the results. For instance, P9: Spains inefficiency of 29% would indicate that the life expectancy could be increased by 29%? If the infant survival rate is increased by 29%, would this mean that more than 1000 infants per 1000 live birth survive?
  •  Also, because it is not clear for which countries the frontier is shaped by many observations, and for which there are few peer units, it is hard for the reader to see the relationship between bias and sample, and bias and explanatory variables.
  • The intro states that countries aim at "the health status of the population, the responsiveness, and equity." Are you able to address these issues based on the results?
  • I am not convinced that the output orientation of the model is appropriate. On the one hand, the output measures are basically outcomes that result from health expenditures in past periods. On the other hand, I suspect that the "health system DMU" has rather control over the expenditures than directly over the outcomes. You somehow state this also in the last paragraph (" he factors cushioning the effect may be of historical,nsocial or cultural nature, and may be associated with both the hierarchical nature of societies, as indicated by income inequality, and with the health of the population. "

Author Response

We appreciate the suggestions of the reviewer that have contributed to the improvement of this research.

Reviewer 2 Report

It is an interesting study focusing on evaluation of the efficiency of European health systems using fuzzy data envelopment analysis. However, there are some issues:

(1) The English quality should be further improved;

(2) The motivations should be clearly described;

(3) The contributions should be listed;

(4) Authors should carefully check all the equations. All the mathematical symbols should be clearly interpreted when they appear the first time;

(5) Some recent studies about data envelopment analysis and kernel density estimation should be included:

Evaluation of startup companies using multicriteria decision making based on hesitant fuzzy linguistic information envelopment analysis models;

Determine OWA operator weights using kernel density estimation;

Assessment and selection of smart agriculture solutions using an information error‐based Pythagorean fuzzy cloud algorithm.

Author Response

(The authors gave the same response as above.)
